

# More on the three-gluon vertex in SU(2) Yang-Mills theory in three and four dimensions

Axel Maas[1][*] and Milan Vujinović[2][†]

Institute of Physics, NAWI Graz, University of Graz,
Universitätsplatz 5, A-8010 Graz, Austria

## Abstract

The three-gluon vertex has been found to be a vital ingredient in non-perturbative functional approaches. We present an updated lattice calculation of it in various kinematical configurations for all tensor structures and multiple lattice parameters in three dimensions, and in a subset of those in four dimensions, for SU(2) Yang-Mills theory in minimal Landau gauge. In three dimensions an unambiguous zero crossing for the tree-level form-factor is established, and consistency for all investigated form factors with a power-like divergence towards the infrared is observed. Using very coarse lattices this is even seen towards momenta as low as about 15 MeV. The results in four dimensions are consistent with such a behavior, but do not yet reach deep enough into the infrared to establish it.


# 1   Introduction

Vertices are the central quantities to encode interactions. Of particular importance among them are the primitively divergent ones. In Yang-Mills theory these are the three-gluon vertex, the ghost-gluon vertex, and the four-gluon vertex. Besides the fact that they encode themselves information on the interactions, they are also the building blocks for solutions of functional equations, like Dyson-Schwinger equations and functional renormalization group equations.

Among the primitively divergent vertices the ghost-gluon vertex showed so far least modification from its tree-level behavior [1–10]. The three-gluon vertex, however, showed quite surprising features, especially at low momenta [1–3,5,6,9–21]. Lattice results find that in two dimensions its tree-level form factor shows unambiguously a zero crossing at about 450(50) MeV momenta on the largest investigated volume together with a power-like divergence towards negative infinity [10]. In three dimensions, results suggested the existence of such a zero-crossing at a few hundred MeV [1, 12]. However, only the lowest non-vanishing momentum point has been found to signal a zero crossing, which is notoriously affected by systematic errors. In four dimensions, it depends on the judgment of statistical and systematic uncertainties, whether a zero crossing has been observed [1, 11, 18, 19, 21]. Functional results [2, 3, 5, 6, 13–17, 20] strongly suggest that the zero crossing is always present, though it is not yet unambiguously decided whether in higher dimension a divergence occurs. The situation for the other tensor structures has been much less investigated, but depending on definitions they also show non-trivial behavior [2,3,5,6,11,13–15,17,21]. The four-gluon vertex is substantially more involved, though results indicate a similar non-trivial behavior [2,3,22].

Here, we will investigate the three-gluon vertex of SU(2) Yang-Mills theory using lattice gauge theory in minimal Landau gauge in three dimensions for all tensor structures in a number of kinematic configurations at high statistics. We establish its zero crossing in the thermodynamic limit at around $400-515$ MeV, depending on momentum configuration, and find substantial evidence in favor of a power-like divergence towards zero momentum. A corresponding cross check in four dimensions, which accounts for the same type of systematic uncertainties, but only for the tree-level tensor structure, is compatible with the qualitative behavior in three dimensions. We establish an upper limit for the zero crossing of $120-240$ MeV, depending again on momentum configuration. This indicates a substantial difference between four dimensions and lower dimensions.

Our setup is briefly discussed in section 2. Results are presented in section 3, for the three-dimensional tree-level form factor in section 3.1, for the non-tree-level form factors in section 3.2, and for the four-dimensional tree-level form factor in section 3.3. We conclude in section 4. As we employ partly very coarse lattices in three dimensions we discuss the corresponding scale setting in appendix A. While the main text only discusses power-law-like behavior a logarithmic behavior has been suggested in four dimensions [2,20,21]. Since a fit of this type turns out to be substantially more unstable than a power-law one, a discussion of it is relegated to appendix B.

# 2   Setup

Our aim is twofold. One is the behavior of the tree-level form factor at very small momenta, especially in three dimensions. As the ghost-gluon vertex exhibited unexpectedly large lattice artifacts in some, but not all, momentum configurations previously investigated [8], an important step is to constrain lattice artifacts. Thus, we used for this the relatively straight-forward approach of [12,23] to determine the vertex function on a number of different lattice settings. The list of lattice setups is listed in table 1. Especially, the results are based on the unimproved Wilson action in minimal Landau gauge.

Table 1: Number and parameters of the configurations used, ordered by dimension, lattice spacing, and physical volume. See [12, 23] for technical details. Autocorrelation times of local observables have been monitored to ensure decorrelation. See [24] and appendix A on details of how the lattice spacing was determined. Config. is the number of configurations. Tensor indicates whether this setting has been used to determine tree-level (tl) or non-tree-level (ntl) form factors.

| $d$ | $N$ | $\beta$ | $a\,[\mathrm{fm}]$ | $a^{-1}\,[\mathrm{GeV}]$ | $L\,[\mathrm{fm}]$ | config. | Tensor |
|---|---|---|---|---|---|---|---|
| 3 | 40 | 0.634 | 0.985 | 0.200 | 39.4 | 120690 | tl |
| 3 | 60 | 0.634 | 0.985 | 0.200 | 59.1 | 145504 | tl |
| 3 | 80 | 0.634 | 0.985 | 0.200 | 78.8 | 202818 | tl |
| 3 | 40 | 0.880 | 0.821 | 0.240 | 32.8 | 120745 | tl |
| 3 | 60 | 0.880 | 0.821 | 0.240 | 49.3 | 143555 | tl |
| 3 | 80 | 0.880 | 0.821 | 0.240 | 65.7 | 151319 | tl |
| 3 | 40 | 1.27 | 0.657 | 0.300 | 26.3 | 85488 | tl |
| 3 | 60 | 1.27 | 0.657 | 0.300 | 39.4 | 154551 | tl |
| 3 | 80 | 1.27 | 0.657 | 0.300 | 52.6 | 99347 | tl |
| 3 | 40 | 1.94 | 0.438 | 0.450 | 17.5 | 85488 | tl |
| 3 | 60 | 1.94 | 0.438 | 0.450 | 26.3 | 99522 | tl |
| 3 | 80 | 1.94 | 0.438 | 0.450 | 35.0 | 115175 | tl |
| 3 | 40 | 3.18 | 0.246 | 0.802 | 9.84 | 102137 | tl |
| 3 | 60 | 3.18 | 0.246 | 0.802 | 14.8 | 109304 | tl |
| 3 | 80 | 3.18 | 0.246 | 0.802 | 19.7 | 111700 | tl |
| 3 | 32 | 3.60 | 0.210 | 0.940 | 6.71 | 33696 | ntl |
| 3 | 48 | 3.60 | 0.210 | 0.940 | 10.1 | 33696 | ntl |
| 3 | 64 | 3.60 | 0.210 | 0.940 | 13.4 | 33696 | ntl |
| 3 | 80 | 3.60 | 0.210 | 0.940 | 16.8 | 33696 | ntl |
| 3 | 32 | 4.00 | 0.184 | 1.07 | 5.89 | 33696 | ntl |
| 3 | 48 | 4.00 | 0.184 | 1.07 | 8.83 | 33696 | ntl |
| 3 | 64 | 4.00 | 0.184 | 1.07 | 11.8 | 33696 | ntl |
| 3 | 80 | 4.00 | 0.184 | 1.07 | 14.7 | 33696 | ntl |
| 3 | 32 | 4.40 | 0.164 | 1.20 | 5.24 | 33696 | ntl |
| 3 | 48 | 4.40 | 0.164 | 1.20 | 7.86 | 33696 | ntl |
| 3 | 64 | 4.40 | 0.164 | 1.20 | 10.5 | 33696 | ntl |
| 3 | 80 | 4.40 | 0.164 | 1.20 | 13.1 | 33696 | ntl |
| 3 | 40 | 5.61 | 0.123 | 1.60 | 4.92 | 86451 | tl |
| 3 | 60 | 5.61 | 0.123 | 1.60 | 7.38 | 86935 | tl |
| 3 | 80 | 5.61 | 0.123 | 1.60 | 9.84 | 81060 | tl |
| 3 | 40 | 10.5 | 0.0616 | 3.21 | 2.46 | 91093 | tl |
| 3 | 60 | 10.5 | 0.0616 | 3.21 | 3.70 | 108724 | tl |
| 3 | 80 | 10.5 | 0.0616 | 3.21 | 4.93 | 111447 | tl |
| 4 | 16 | 2.1306 | 0.246 | 0.800 | 3.94 | 63772 | tl |
| 4 | 24 | 2.1306 | 0.246 | 0.800 | 5.90 | 49600 | tl |
| 4 | 32 | 2.1306 | 0.246 | 0.800 | 7.87 | 44671 | tl |
| 4 | 16 | 2.3936 | 0.123 | 1.60 | 1.97 | 78668 | tl |
| 4 | 24 | 2.3936 | 0.123 | 1.60 | 2.95 | 70703 | tl |
| 4 | 32 | 2.3936 | 0.123 | 1.60 | 3.94 | 35256 | tl |
| | | | | | | Continued on next page | |

| Table 1 continued | | | | | | | |
|---|---|---|---|---|---|---|---|
| $d$ | $N$ | $\beta$ | $a$ [fm] | $a^{-1}$ [GeV] | L [fm] | config. | Tensor |
| 4 | 16 | 2.5977 | 0.0616 | 3.20 | 0.986 | 50940 | tl |
| 4 | 24 | 2.5977 | 0.0616 | 3.20 | 1.48 | 65792 | tl |
| 4 | 32 | 2.5977 | 0.0616 | 3.20 | 1.96 | 37327 | tl |

The (unrenormalized) form factors $\Gamma^j$ are generally obtained [12] by projection and amputation as

$$\Gamma^j = \frac{\Gamma^{0j}_{\mu\nu\rho abc} \left\langle A^a_\mu A^b_\nu A^c_\nu \right\rangle}{\Gamma^{0j}_{\alpha\beta\gamma def} D^{dg}_{\alpha\sigma} D^{eh}_{\beta\omega} D^{fi}_{\gamma\delta} \Gamma^{0j}_{\sigma\omega\delta ghi}} \,.$$

Herein the $D$ are the gluon propagators. They can be found in [8] and appendix A for the range of lattice settings used here. Errors are purely statistical[1].

The $\Gamma^{0j}$ are suitable base tensors, of which four transverse ones are sufficient to determine the three-gluon vertex in Landau gauge completely [27]. Thus, the form factors $\Gamma^j$ yield the deviations from the base tensors $\Gamma^{0j}$, where any constant prefactors can be absorbed judiciously into these base tensors. Here, the base tensor $\Gamma^{00}$ will be the lattice tree-level tensor from [28], as was used in [12].

For the remaining three base tensors, we use the Bose-symmetric basis developed in [17, 23]. For the non-tree-level vertices we are primarily interested in an exploration of their low-momentum behavior. Due to asymptotic freedom, they should approach zero at sufficiently large momenta in three dimensions[2]. Given the complexity in deriving the non-tree-level lattice form factors at leading perturbative order [28], which to our knowledge has not yet been calculated in three dimensions, we employ their continuum versions [17] instead, and only determine them up to $ap < 1$. It turns out that this is not a serious limitation, as above this momenta all of them are found for all considered momentum configurations to be consistent with zero within errors.

We consider thus the following three additional tensor structures [17]

$$\Gamma^{01}_{\mu\nu\rho}(p_1, p_2, p_3) = t^1_\mu t^2_\nu t^3_\rho, \tag{1}$$

$$\Gamma^{02}_{\mu\nu\rho}(p_1, p_2, p_3) = p^2_1 t^1_\mu \delta_{\nu\rho} + p^2_2 t^2_\nu \delta_{\rho\mu} + p^2_3 t^3_\rho \delta_{\mu\nu}, \tag{2}$$

$$\Gamma^{03}_{\mu\nu\rho}(p_1, p_2, p_3) = \omega_1 t^1_\mu \delta_{\nu\rho} + \omega_2 t^2_\nu \delta_{\rho\mu} + \omega_3 t^3_\rho \delta_{\mu\nu}, \tag{3}$$

$$t^1 = p_2 - p_3,$$
$$t^2 = p_3 - p_1,$$
$$t^3 = p_1 - p_2,$$
$$\omega_1 = p^2_3 - p^2_2,$$
$$\omega_2 = p^2_1 - p^2_3,$$
$$\omega_3 = p^2_2 - p^2_1,$$

using unimproved lattice momenta [23]. Note that for SU(2) the only color structure is the totally anti-symmetric Levi-Civita tensor, which is therefore suppressed. In contrast to the tree-level vertex, all of them have a larger mass dimension, due to the different momentum structure. This is taken care of when isolating the dimensionless form factors.

---

[1]Unfortunately, the errors are, as usual, exponentially larger than for the gluon propagator due to the larger number of operators [25]. Techniques like smearing to reduce the noise does alter the behavior of momentum-resolved quantities qualitatively [26] and thus cannot be used.

[2]Note that there may be potentially logarithmic deviations, due to the fact that non-perturbative resummation in three dimensions does yield non-trivial additional corrections [29].

As in [12] introduced, we will consider three different momentum configurations. One is the symmetric one, in which all momenta are of equal magnitude. The second is the back-to-back configuration, in which one momentum vanishes. The third has two momenta at 90 degrees, but otherwise unconstrained. This last one is only considered for the tree-level form factor. Note that the form factors are all symmetric in all arguments, like the vertex itself. The choice of basis (1-3) has the consequence that for the symmetric configuration all but the $\Gamma^0$ and $\Gamma^1$ form factors vanish for kinematical reasons [17].

## 3 Results

### 3.1 Tree-level form-factor in three dimensions

The results for the tree-level form factor in three dimensions are shown in figure 1. It is visible that at large momenta the form factor behaves as expected, and approaches one quickly above roughly 2 GeV. It is also visible that the form factor drops below zero around 300-500 MeV, relatively independent of the lattice parameters. However, lattice artifacts play a relevant role at low momenta. This is visible especially in the back-to-back configuration, as was also observed for the ghost-gluon vertex [8]. In fact, the lowest momentum point is entirely dominated by lattice artifacts, and is excluded. It is visible that an increase in volume at fixed discretization tends to make the form factor more negative, while a finer discretization at fixed volume tends to do the opposite. As a consequence, the results on the coarser lattices are systematically below the ones on the finer lattices in the infrared. This effect is statistically significant, and not a small effect. It is also visible, especially again for the back-to-back configuration, that the largest momenta are dominated by lattice artifacts.

We fitted each lattice setup by two ansätze

$$\Gamma_1(p^2) \;=\; 1 - a\,(p)^{-b}\,, \tag{4}$$

$$\Gamma_2(p^2) \;=\; 1 - \frac{a^2}{p^2 + m^2}\,, \tag{5}$$

i. e. by a power-law ansatz and a dominant-pole ansatz, allowing explicitly for zero mass. The latter would be expected in a perturbative setting. None of the lattice setups show explicitly a flattening at small momenta, and thus we did not include a fit with a crossing of the zero momentum axis with flat slope.

We did the fits explicitly for the back-to-back momentum configuration and the symmetric momentum configuration. We included all points below $ap = 1$. The fits become harder and harder on smaller volumes, as there is less and less distinction from the asymptotic constant behavior. We did not obtain reasonable fit results for the dominant-pole ansatz (5), but achieved very good fit-quality, with $\chi^2$ values[3] below 2 in most cases, with the power-law ansatz (4). This is strikingly different from the situation in presence of a Brout-Englert-Higgs effect [30], where (5) works excellent for the three-gluon vertex. We concentrate therefore in the following on the power-law ansatz (4). The fit is also shown in figure 1. It is also visible, by comparing the fit to a slightly different momentum configuration, that the parameters are angle-dependent.

The asymptotic behavior is emphasized by plotting

$$\ln(1 - |\Gamma^0|), \tag{6}$$

rather than the dressing function itself, against $\ln p$, as putting (4) into (6) yields a straight-line for a power-law behavior. This is indeed the case, as is visible in figure 2.

---

[3]Fitting using (6) below implies a linear fit, and thus the $\chi^2$ value is meaningful.

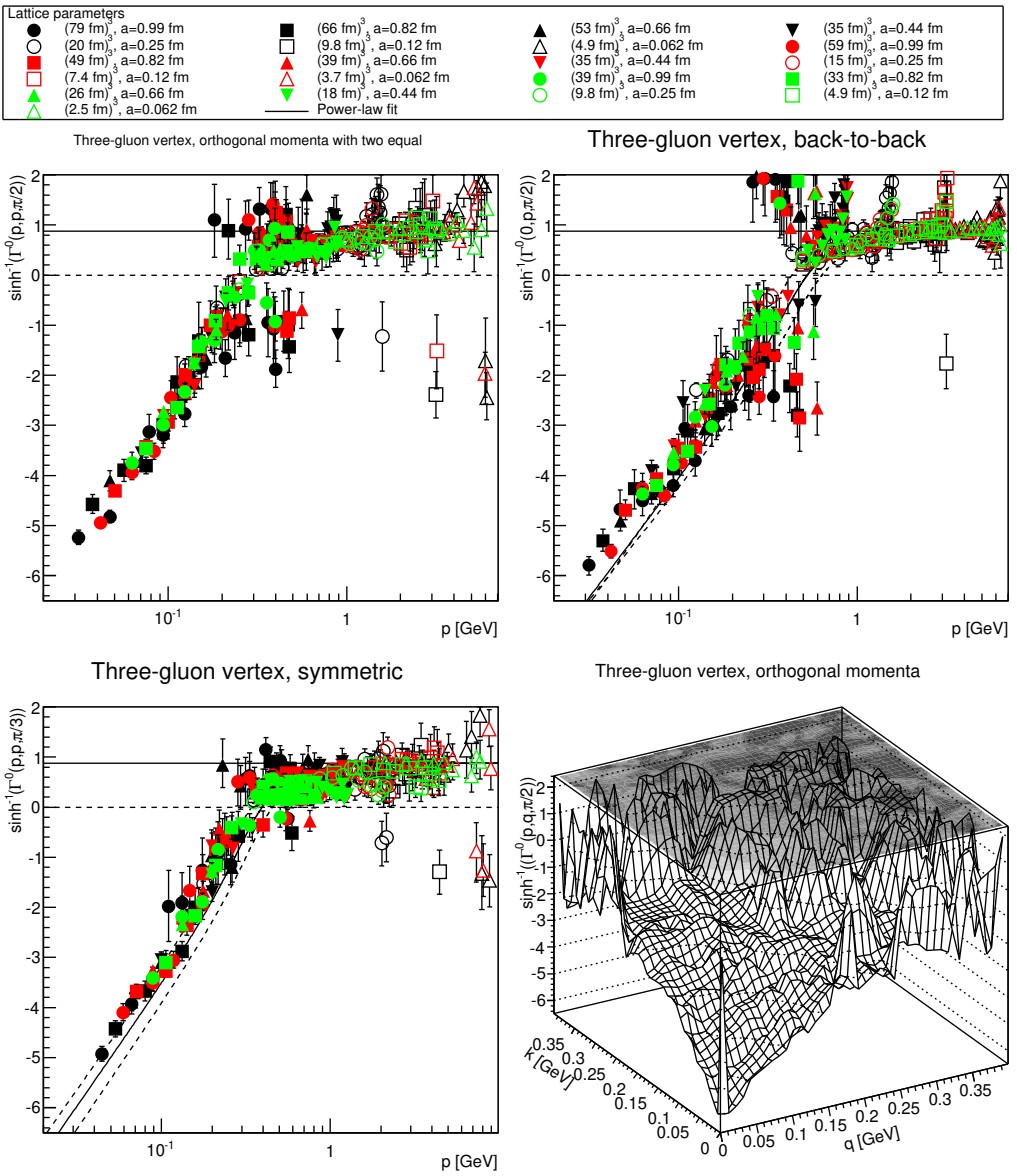

Figure 1: The three-gluon vertex tree-level form factor in three dimensions. The top-left panel shows a cut along the diagonal of the lower-right plot. The latter shows the situation with two momenta being orthogonal to each other, with interpolation of the data points for the largest volume at the coarsest discretization. The top-right panel shows the back-to-back momentum configuration and the bottom-left panel the symmetric momentum configuration. Only points with a relative error of less than 100% are shown, and the lowest momentum point is suppressed. The fit (4) shown is the extrapolated one in the symmetric configuration (bottom-left panel) and in the back-to-back configuration (top-right panel), see text.

To allow for a continuum extrapolation, the fits have been done for all lattice settings. The results are shown in figure 3. The prefactor shows little dependence on the lattice parameters within uncertainty, but depends on the momentum configuration. This emphasizes a slight angular dependency of the form factor, as seen already in figures 1 and 2. The exponent is substantially volume-dependent, and in different ways for the different kinematic configurations. It has, within errors, also not reached its infinite-volume behavior on the current lattice

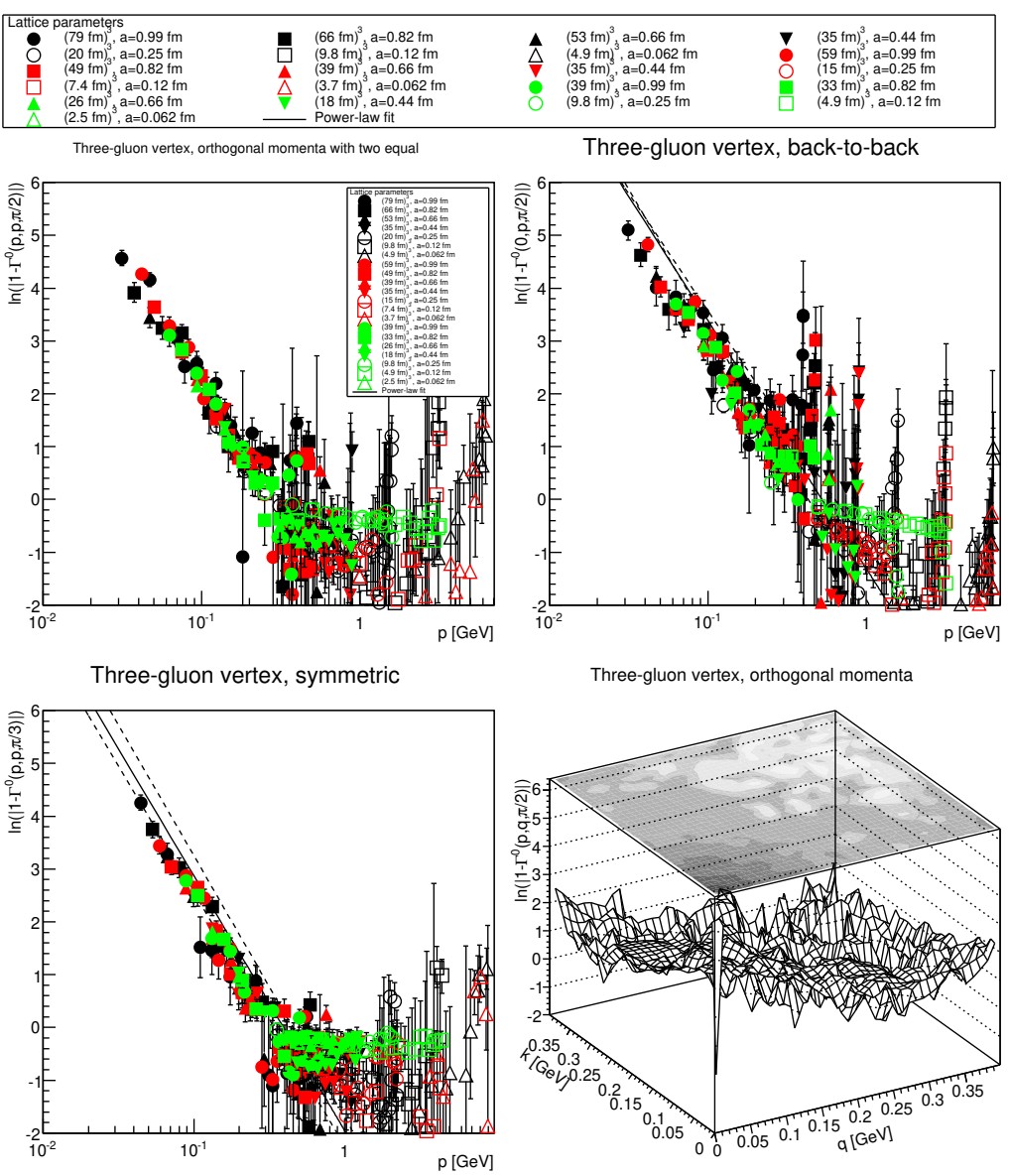

Figure 2: The three-gluon vertex tree-level form factor in three dimensions plotted as $\ln|1-\Gamma^0|$ The top-left panel shows a cut along the diagonal of the lower-right plot. The latter shows the situation with two momenta being orthogonal to each other, with interpolation of the data points for the largest volume at the coarsest discretization. The top-right panel shows the back-to-back momentum configuration and the bottom-left panel the symmetric momentum configuration. Only points with a relative error of less than 100% are shown, and the lowest momentum point is suppressed. The fit (4) shown is the extrapolated one in the symmetric configuration (bottom-left panel) and in the back-to-back configuration (top-right panel), see text.

settings. The infinite-volume-extrapolated results for the back-to-back configuration and the symmetric configuration are

$$\Gamma_b = 1 - 0.25^{+15}_{-11} p^{-2.04^{+22}_{-13}},$$
$$\Gamma_s = 1 - 0.14(4) p^{-2.1(1)},$$

respectively. These extrapolated values also confirm the slight angular dependence. Note that

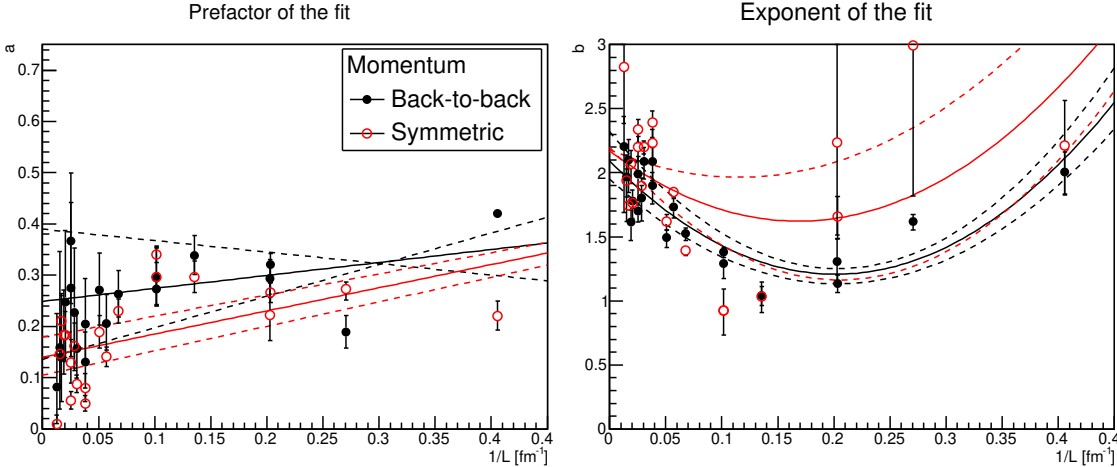

Figure 3: The fit parameters of (4) in 3 dimensions, together with a fit for the volume dependence. For the prefactor the fit is linear in the inverse lattice extent, and quadratic for the exponent.

the value of the exponent is within errors compatible with two, and would thus be compatible with a pure massless pole behavior. It is below the prediction of a scaling behavior [31], which is between 2.4 and 2.6 in the symmetric configuration. In addition, this yields a zero crossing at $399^{+56}_{-47}$ MeV in the symmetric configuration and $515^{+104}_{-90}$ MeV in the back-to-back configuration in the thermodynamic limit, slightly above the one on the largest volumes employed here.

### 3.2 Non-tree-level form-factors in three dimensions

The results for the non-tree-level form factors are shown in figure 4. Only in the symmetric case, where only one of them is potentially non-vanishing, any statistically substantial deviations from zero is visible. In the orthogonal case all non-tree-level form factors are within errors consistent with zero. However, this needs to be interpreted as upper limits only. Still, even if they are non-zero, they are substantially smaller than the tree-level form factor. This is in good agreement with results from functional studies [2, 3, 5, 6, 13–17].

The result in the symmetric case is still marginally compatible with a non-zero and negative value below roughly 500 MeV. It is still much smaller than the tree-level form factor. Above this momentum, it is again too small to be resolved by the current statistics. The same form-factor shows in the back-to-back configuration an upward fluctuation below 500 MeV momentum.

Given the size of the fluctuations, these results strongly suggest that substantially more than an order of magnitude of statistics is necessary to have a chance of seeing a signal for any statistical sound non-zero behavior in the back-to-back case. This is also necessary in the symmetric case to obtain a statistically significant sound signal for a deviation from zero in the infrared. Thus, this needs to be postponed until such an amount of computing time becomes available for this purpose.

However, the behavior seen is indeed consistent with the same power-law behavior as for the tree-level. This is indicated by the fits using the same exponents as in figure 1, and only adjusting the prefactor to -0.01 in the symmetric case and to 0.008, -0.02, and 0.015 in the back-to-back case for the three form factors. The only other change was to drop the tree-level one. This suggests that the qualitative behavior of all form factors are the same within the present statistical uncertainty, when the tree-level behavior is taken into account.

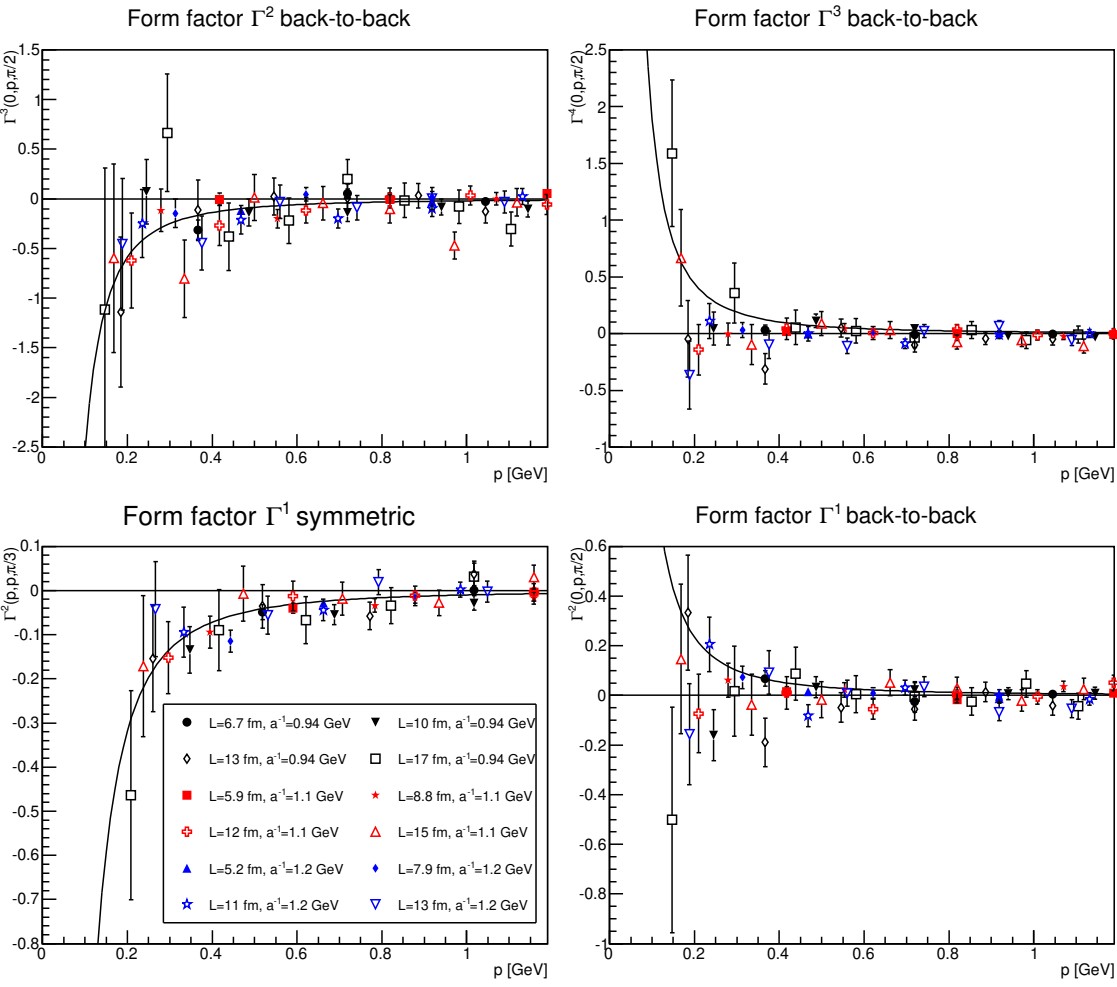

Figure 4: The non-tree-level form-factor of three-gluon vertex in three dimensions for the investigated momentum configurations. The symmetric case with only one non-trivially non-vanishing form factor is shown in the lower-left panel. The remainder are the three non-tree-level form factors in the back-to-back momentum configuration. The fits are discussed in the text.

## 3.3 Tree-level form-factor in four dimensions

The four-dimensional case is much harder to assess, as the higher computational costs make it much harder to obtain the corresponding high statistics on large volumes. Still, it is possible to see whether the deviation from one towards small momenta can be as well described by the power-law fit (4) as in three dimension. However, because of the smaller number of lattice points, the same cuts on including points are too severe. We included all points below 1 GeV momentum as well as all other points with less than 25% relative error in the fits. As an alternative a logarithmic departure, as suggested e. g. in [2, 20, 21], is analyzed in appendix B, but turns out not to work as well.

The results for the tree-level form factor are shown in figure 5. There are two distinct behaviors.

One is in the ultraviolet. There, a distinct dependency on the lattice spacing is observable in the back-to-back configuration, which is not present in the other momentum configurations. This has also been visible in three dimensions in figure 1. A similar observation of pronounced discretization-dependency was also made for the ghost-gluon vertex [8] and

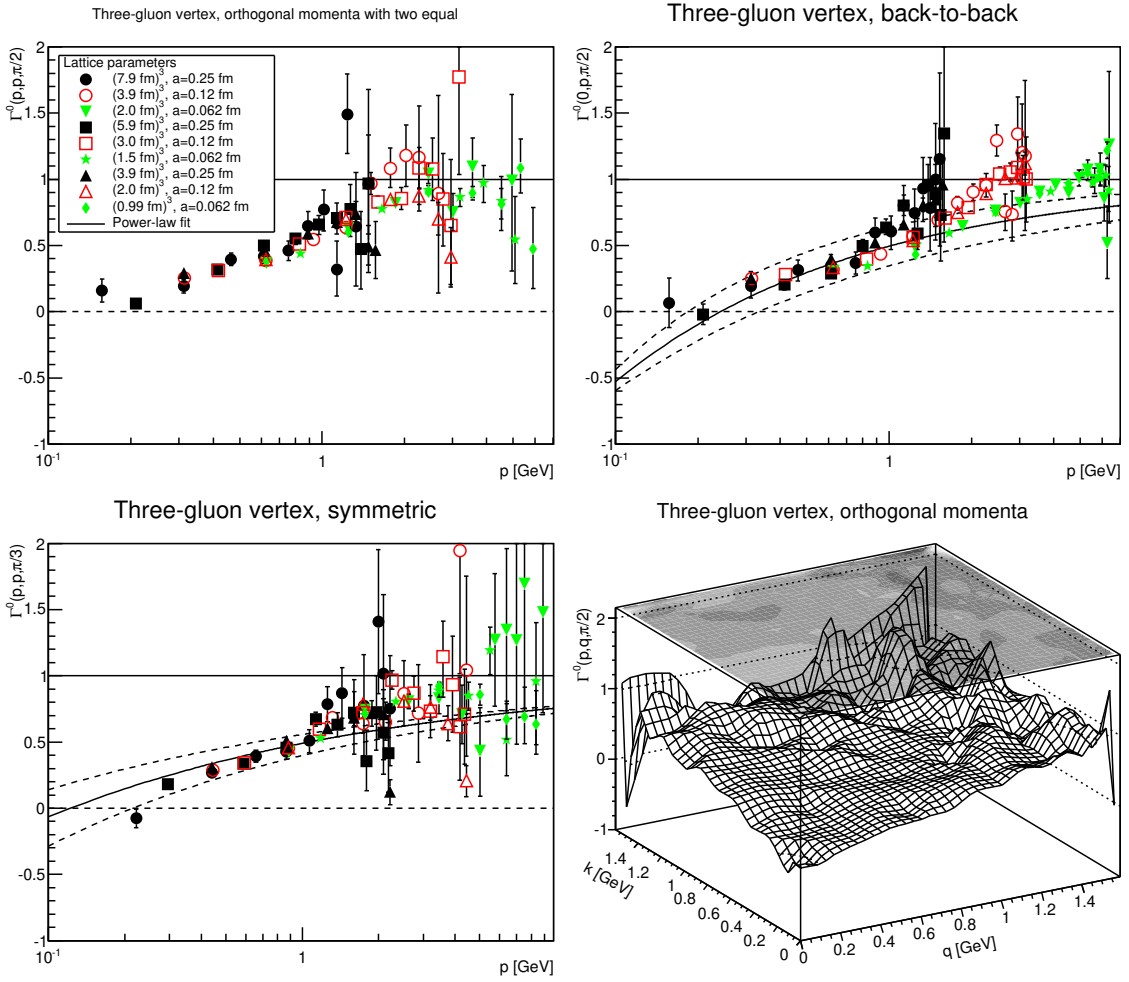

Figure 5: Same as in figure 1, but in four dimensions and the lowest momentum point is included.

(quenched) scalar-gluon vertices [32]. It therefore appears to be a rather general feature of the back-to-back configuration. Therefore, this momentum configuration is not as suitable to extract ultraviolet properties than as, say, the symmetric momentum configuration. We also note that the results are within errors at very large momenta indistinguishable, and thus all renormalization effects below the statistical noise.

As is visible in figure 5 no statistically significant zero-crossing is observed. The infrared behavior is again better emphasized by the transformed form (6). The behavior towards small momenta, as is visible in figure 6, is indeed consistent with a power-law-like deviation from one, as described by (4). Again, the results exhibit a slight angular dependency.

Performing the fits yields the fit parameters shown in figure 7. Again no pronounced dependency on the discretization is observed. The infinite-volume limits of the fits are also again different for different angles, and are found to be

$$\Gamma_b = 1 - 0.51(15)p^{-0.48(11)},$$
$$\Gamma_s = 1 - 0.51(9)p^{-0.32^{+1}_{-4}},$$

which correspond to an expected zero crossing at $122^{+92}_{-63}$ MeV and $242^{+92}_{-59}$ MeV for the back-to-back momentum configuration and the symmetric momentum configuration, respectively. In these momenta ranges the results in figure 5 are indeed compatible with zero within statistical

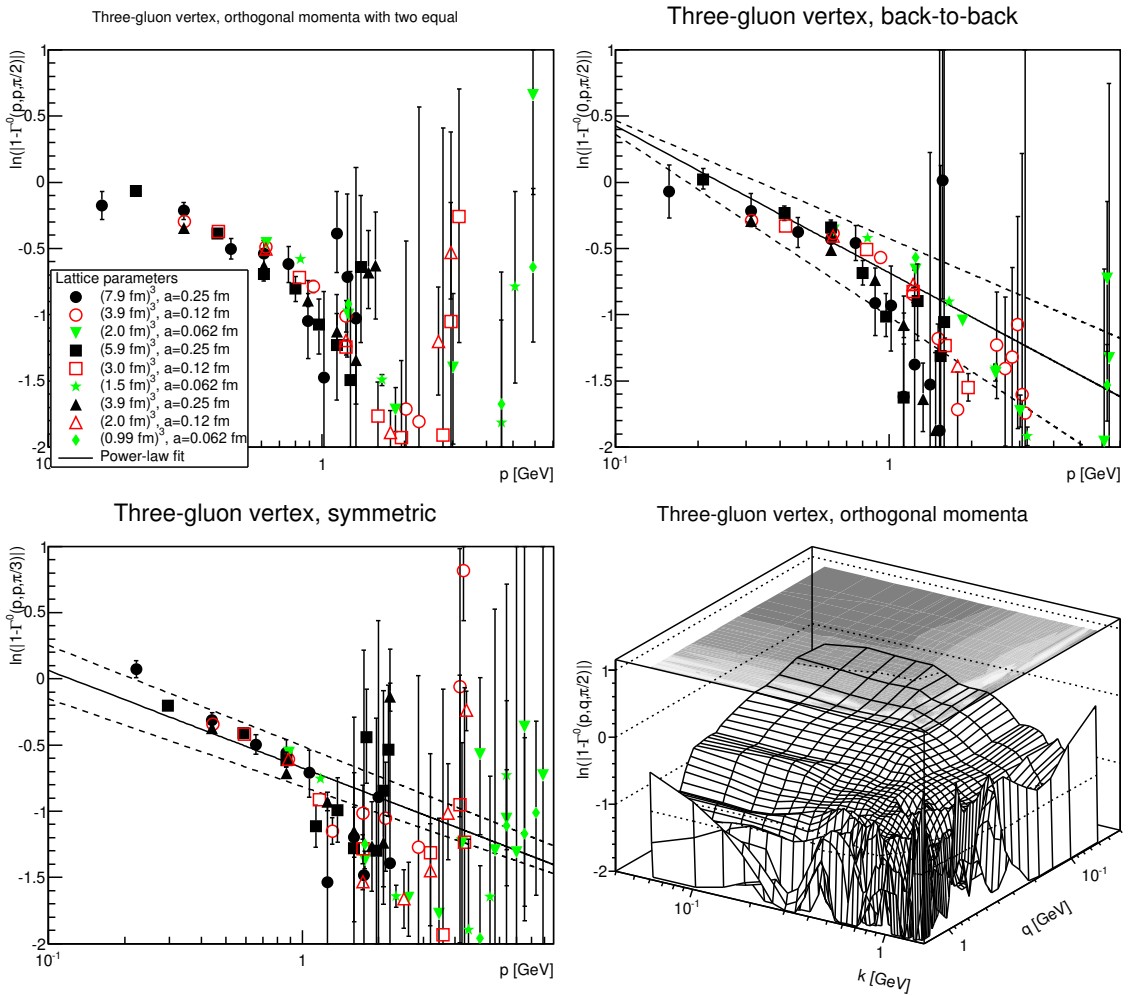

Figure 6: Same as in figure 2, but in four dimensions and the lowest momentum point is included.

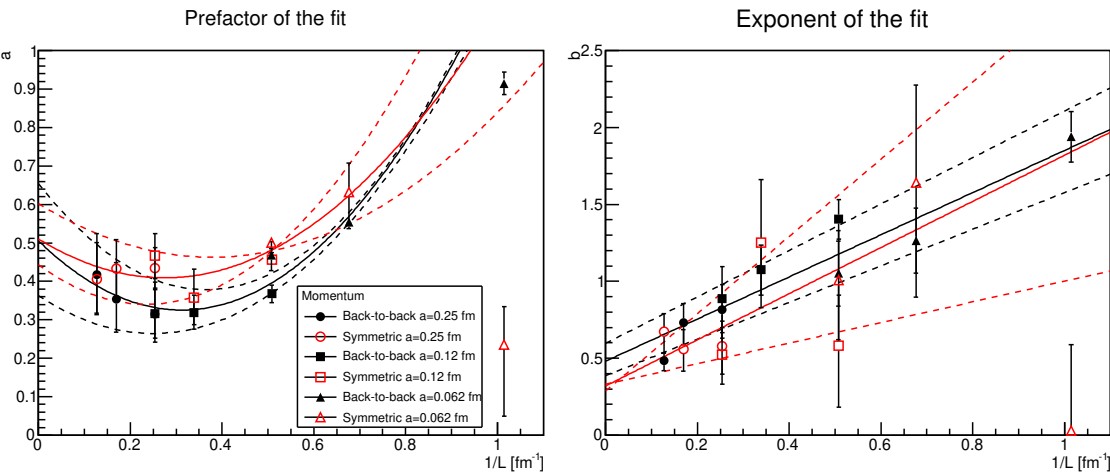

Figure 7: The fit parameters of (4) in 4 dimensions, together with a fit for the volume dependence. For the prefactor the fit is quadratic in the inverse lattice extent, and linear for the exponent. The smallest volume was dropped in the fit.

uncertainty. Hence, the four dimensional form factor shows qualitatively the same behavior as in three dimensions, though with a substantially smaller exponent. This is in as far interesting, as this implies a decrease of the exponent from about 2.2 to roughly 2.1 and 0.4 from two to four dimensions, crossing somewhere the expected behavior for a massless pole. These findings are consistent with those for SU(3) Yang-Mills theory and QCD [11, 33] and other results [2, 3, 5, 6, 13–17, 19].

# 4 Conclusions

In summary, we presented the most comprehensive study of the three-gluon vertex in three dimensions to date. We established an unambiguous zero crossing of the tree-level form factor and find substantial evidence in favor of an infrared divergence with a strength of roughly the one expected from a massless pole. We also presented the first investigation of non-tree-level form factors in three dimensions, and find them to be of negligible size in comparison to the tree-level form factor above roughly 500 MeV. Towards the infrared, we find hints of a similar qualitative behavior as for the tree-level form factor, though with an order of magnitude suppression.

In addition we find evidence that in four dimensions a similar qualitative dependency prevails, though with a much weaker infrared divergence. Consequently, we do not yet reach deep enough into the infrared to establish a zero crossing unambiguously, but could establish the momentum range in which this should occur. As the infrared turns out to be little affected by discretization this suggests that relatively coarse lattices of order $48^4$ with high statistics should be able to establish a zero crossing reliably, if it is indeed there. This is, however, beyond our current capacity.

We find therefore evidence that the critical infrared exponent decreases with dimensionality. The actual strength, i. e. prefactor, is however remarkably similar in all dimensions. We also find evidence for a slight quantitative angle-dependency of the form factors. Also, we find that the back-to-back configuration suffers, as with all other investigated vertices so far, from substantial discretization artifacts in the ultraviolet. This suggests to use other momentum configurations exclusively to investigate the high-momentum features, like anomalous dimensions.

# Acknowledgments

M. V. was supported by the FWF under grant number J3854. The computations have been performed on the HPC clusters at the University of Graz and we are grateful for its fine operation.

# A   Scale setting

The $\beta$ values employed in three dimensions are partially substantially below the range of validity of the interpolation formula for scale setting from [34]. It was therefore necessary to fix the scale in an alternative way. In addition, it is known that at $\beta = 0$ the behavior of correlation functions is qualitatively different [35–37], and moreover there exists some hints for a bulk transition at finite $\beta$ [38]. Thus, the question is non-trivial down to which values of $\beta$ the development is sufficiently smooth for an investigation of the infrared properties of the vertex.

As the vertex requires only gluonic input, we required a smooth behavior of the gluon prop-

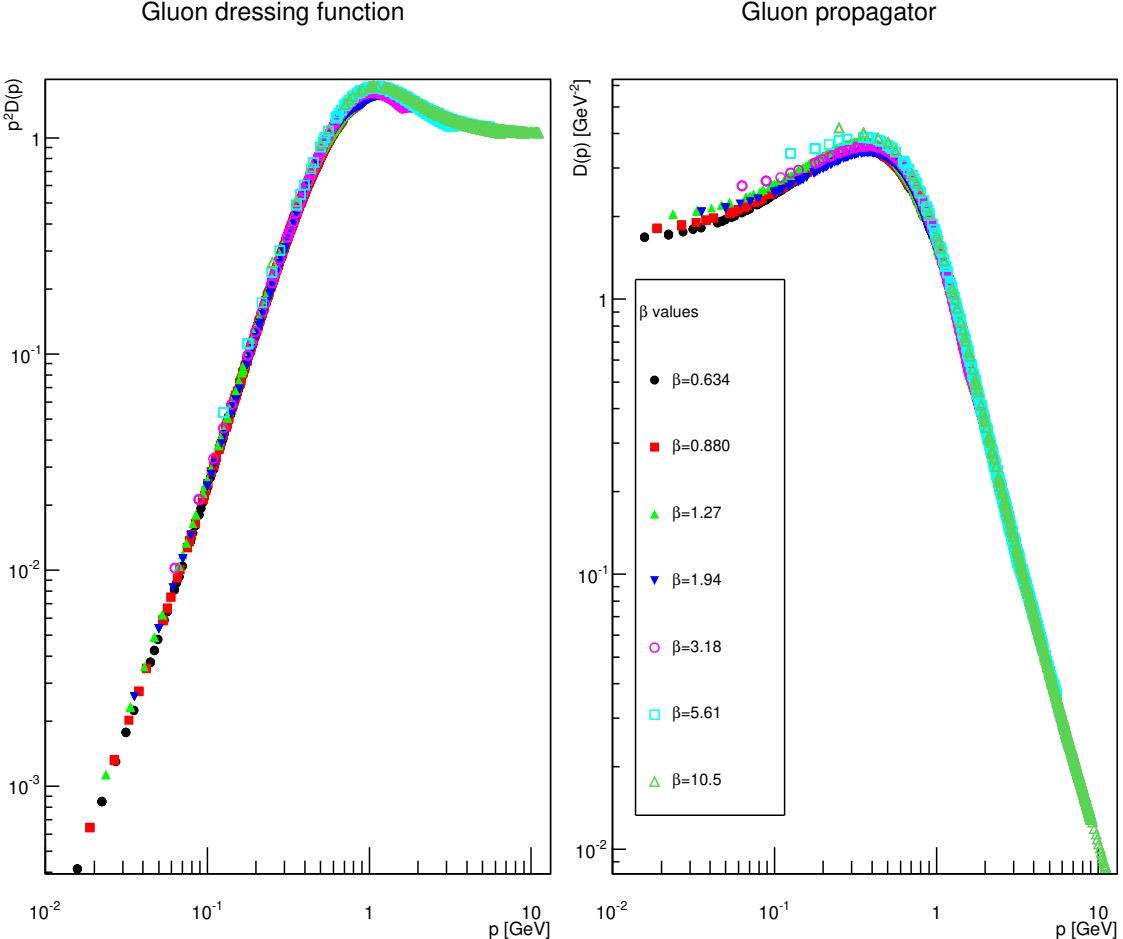

Figure 8: The gluon dressing function (left panel) and the gluon propagator (right panel) for the $80^3$ lattice volumes for all $\beta$ values, using the lattice spacing in table 1.

agator to fix the lattice spacing. This is indeed possible down to the lowest $\beta$-value possible, leading to the values given in table 1. The result is shown in figure 8. Over three orders of magnitudes in momenta, the lowest being just about 15 MeV, the gluon propagator shows no more changes than expected from the change in physical volume in three dimensions [24]. At the same time the dressing function itself changes by almost four orders of magnitude. This strongly suggests that the chosen lattice spacing are, at least, roughly of the correct size and even for the smallest values of $\beta$ not yet the strong-coupling regime has been entered.

# B  Alternative fits

It was suggested that the infrared divergence in four dimensions may be logarithmic [2,20,21]. This would entail a very slow departure from a straight line in figure 6. Also, this result clearly shows a power-law contribution, at least within the momentum range. However, this may only be an approximate behavior. Especially as the 3-dimensional results do not suggest a change of behavior in the very deep infrared.

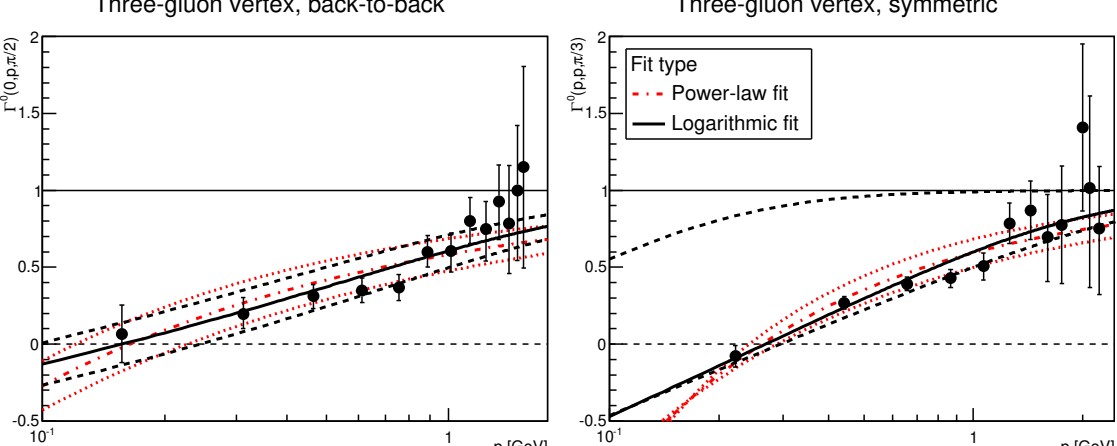

Figure 9: Power-law fits and logarithmic fits compared to the data for the four-dimensional case with the largest physical volume.

Therefore, the data are fitted with the ansatz

$$\Gamma_3(p^2) = 1 - a \left| \ln \frac{p^2}{p^2 + b^2} \right|^c ,$$

where $b$ only takes the role to damp out the logarithmic behavior in the far ultraviolet. Otherwise, the fits are done as before. However, the small number of points and one more available fit parameter allows for a more flexible fit than with the two-parameter power-law fit. Performing the same infinite-volume analysis yields for the back-to-back configuration

$$
\begin{aligned}
a &= 0.38^{+5}_{-10}, \\
b &= 0.3^{+10}_{-2} \text{ GeV}, \\
c &= 0.63^{+15}_{-1},
\end{aligned}
$$

and in the symmetric configuration

$$
\begin{aligned}
a &= -1.3(4), \\
b &= -0.1^{+19}_{-1} \text{ GeV}, \\
c &= 0.4^{+3}_{-1}.
\end{aligned}
$$

The instability of the symmetric case is due to the fact that forcing such a very slow deviation from one yields very soft constraints at smaller physical volumes and consequentially large minimal momenta, yielding large uncertainties.

The situation is illustrated in figure 9 for the largest physical volumes. In the back-to-back case the results indeed show quite a similar behavior. This is strongly different in the symmetric case, which is less affected by lattice artifacts. Here, the slower approach to one at large momenta provides a relatively good handle to constrain a power-law behavior, given the errors. The logarithmic behavior, however, is too slow to be strongly constrained within the error band under the same conditions. It is thus very sensitive to the ultraviolet data as well. It is of similar sensitivity at low momenta as the power-law fit. However, here the almost flat behavior is just so to catch the lowest momentum data point, which is, however, also the one most affected by finite-volume effects. As has been seen in the main text, this point usually overshoots the actual value, and while this only yields a small change for the power-law behavior, this substantially needs to bend the logarithmic behavior.

Summarizing, a reliable check whether a logarithmic behavior instead of a power-alaw behavior would prevail needs thus not only good data at low momenta, but also at high momenta, and especially cannot be drawn in the back-to-back momentum configuration. The power-law behavior is much less sensitive. In addition, if the logarithmic behavior should indeed ultimately prevail, this would yield that the exponent of an attempted power-law fit would tend to become smaller and smaller with increasing physical volume. Such a behavior is also not supported by the data seen in figure 7, which tends to a substantially finite value. In addition, the three-dimensional results strongly suggest that the behavior will not change qualitatively very deep into the infrared. Still, in the end, fits can never reliably exclude a possibility.

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
