# Peer review of "More on the three-gluon vertex in SU(2) Yang-Mills theory in three and four dimensions"

_SciPost Physics Core, doi:SciPost Phys. Core 5, 019 (2022)_

## Round 2 · Referee Report · Anonymous (Referee 1) · 2020-9-24

Report

The authors study the 3-gluon correlation functions of Yang-Mills theory in the Landau gauge with lattice simulations. The problem has been studied in the past and the results presented in this article correspond to a improvement on older calculations. The main interest of this article to my opinion is to strengthen the case in favor of a "zero crossing" of the 3-point correlation (ie, the correlation function changes sign at some infrared momentum) in the three-dimensional case. The physically more interesting four-dimensional situation is more challenging and remains open. To my opinion the results are interesting but do not correspond to the criteria of scipost physics for publication (no "groundbreaking computational dicovery"). As such, I think that the article should not be published in scipost physics. A transfer to scipost core could be considered.

1) Apart from showing evidence in favor of the zero crossing, the authors try to characterize the infrared behavior of the correlation function. They fit their lattice results with a power law, eq. (4), and try to extract the associated exponent b. I must say that I am not convinced by the determination of the error bars, which seem underestimated. For instance, in section 3-1, the authors give a determination of the critical exponent b as 1.76 with an error of a few percents. But the lattice data show an approximate linear behavior in the log-log plot on only 1 order of magnitude (see fig 2). On such a small range of momenta, the determination of the exponent with this precision would need to quantify the effect of corrections to scaling and this is not addressed by the authors. (A small comment: on the top left plot of fig. 2 The fit seems to be always above the lattice points. This looks strange.)

2) The authors seem to ignore the calculation of Pelaez et al (arXiv:1310.2594) where the 3-point correlation function was computed. These authors proposed a simple understanding for the zero crossing and give a prediction for the singular behavior of the correlation function, both in 3 and 4 dimensions. The authors should compare their findings with earlier predictions for their exponent b.

3) The article is in general well written. I should mention though that:

-below eq 3, I didn't understand what the author meant in the sentence: "Note that for SU(2) ... suppressed";

-in the next sentence it is not clear to me what "all of them" corresponds to;

-next paragraph, the first sentence should be rewritten.

---

## Round 2 · Referee Report · Anonymous (Referee 2) · 2020-10-12

Report

Report on the manuscript “More on the three-gluon vertex in SU(2) Yang-Mills theory in three and four dimensions”.

Dear Editors,

I read carefully the manuscript. The Landau gauge 3-gluon vertex is analyzed with a Monte-Carlo lattice calculation. Both the three and the four dimensional cases are considered. Tree-level and non-tree-level tensor structures are considered. The authors are able to discuss with more precision this vertex and, in particular, to determine more reliably the existence of a “zero-crossing” in three-dimension (also suggesting a similar but weaker effect in d=4). The article is interesting, address an active topic and is, on general grounds, well-written. However, in my opinion, as it stands it is not suited for publication in SciPost. Moreover, it is not clear to me if the results require its publication in a review with high impact as SciPost because the results are, essentially, an improvement of previous ones on the same topic (some of them from one of the authors of the present manuscript). Even if improving the precision (and extending the results to new tensorial structures) is welcome and should be published, maybe SciPost is not the proper place.

On top of this issue (that is more an editorial choice), in my opinion, the manuscript has many points that should be improved and could be done easily.

1) On the bibliographic side, in my opinion, the quoted literature is a little bit biased. For example, the Peláez et al. article on 3-gluon vertex which is a very well-known reference on the topic is not quoted. This is not just a reference problem because it may also bias the choice of fitting functions. For example, in the Peláez paper it is proposed that the infrared behavior of the 3-gluon vertex is logarithmic in momenta and not a power-law in d=4 (giving a very simple explanation for that). This is qualitatively consistent to what is seen in the present manuscript where it is seen that the apparent infrared divergence is weaker in d=4 than in lower dimensions but no discussion on this possibility is done. Even more, no fit with respect to a logarithmic function is done in d=4 that could be included very easily. On top of that, in the literature concrete values for the infrared exponents are proposed for d=3 in many articles (including Peláez’s) and no concrete comparison to those values is done. I think that this is very easy to be incorporated in the present manuscript.

2) In my impression, the first equation (without number, before (1)) defines a vertex which is not properly renormalized. At odds with the ghost-gluon vertex that a similar expression gives a vertex which is already renormalized, for the 3-gluon vertex the considered expression would have in d=4 an ultraviolet multiplicative divergence. This is not problematic for the purposes of the present article but should be pointed out in the text in order to avoid any confusion.

3) Two lines after equation (5), the authors mention perturbative results to compare to but they do not include any reference. A reference would be welcome.

4) When estimating power-law exponents one usually requires at least a range of two orders of magnitude of variation of the independent variable. Here the authors present figures as 2 and 6 with only one order of magnitude in the power-law range of momenta. In my opinion, this short range of momenta permits to suggest that a power law is present but does not allow to calculate the exponents beyond the order of magnitude. In particular, in my opinion, one cannot estimate with such a narrow interval of momenta exponents with the number of digits that the authors claim. My opinion seems to be confirmed by the figures 3 and 7. Moreover, the authors observe an important volume dependence of their results. In my opinion, the precision of the exponents is overestimated by at least an order of magnitude. I think a rephrasing and a much more careful wording on this point would be welcome. On the same token, the authors suggest that the power-law exponents for the non-tree-level tensor structures are the same than those for the tree-level one. In my opinion, a much careful wording on this point would be welcome because I do not see how with these data one can claim that (beyond a qualitative assertion).

5) In figure 5 one observes a very important noise in the ultraviolet. This has been observed for the same quantity in previous lattice simulations. In my opinion, the reason for such large fluctuations should be discussed in the manuscript or at least the point commented. It may happen that the authors do not know the reason that provokes such a noisy curve but at least they must mention a list of possible origins and discuss more properly the point (that is just mentioned in the manuscript).

6) I do not understand the first sentence of page 5.

I think that those points are easily taken into account. If that were done, in my opinion, the manuscript would be suited for publication. In my opinion the content cannot be qualified as “groundbreaking” or similar qualifications as is expected for SciPost but I think that this is more an editorial decision than a referral decision. After corrections the manuscript could be accepted in an alternative journal.

---

## Round 3 · Author Response

Dear editor and referee,

thank you for the reports. We will discuss below that the creation of additional data has delayed our response somewhat, as well as Corona substantially, in order to address some of the concerns of the referees.

Concerning a shift from SciPost Physics to SciPost Physics Core, we do not have a strong opinion, and a re fine with a shift.

---

## Round 3 · List of Changes

Warnings issued while processing user-supplied markup:

  • Inconsistency: Markdown and reStructuredText syntaxes are mixed. Markdown will be used.
    Add "#coerce:reST" or "#coerce:plain" as the first line of your text to force reStructuredText or no markup.
    You may also contact the helpdesk if the formatting is incorrect and you are unable to edit your text.

We reply now in detail to the raised points of the referees:

Report 1

1) On the bibliographic side, in my opinion, the quoted literature is a little bit biased. For example, the Peláez et al. article on 3-gluon vertex which is a very well- known reference on the topic is not quoted. This is not just a reference problem because it may also bias the choice of fitting functions. For example, in the Peláez paper it is proposed that the infrared behavior of the 3-gluon vertex is logarithmic in momenta and not a power-law in d=4 (giving a very simple explanation for that). This is qualitatively consistent to what is seen in the present manuscript where it is seen that the apparent infrared divergence is weaker in d=4 than in lower dimensions but no discussion on this possibility is done. Even more, no fit with respect to a logarithmic function is done in d=4 that could be included very easily. On top of that, in the literature concrete values for the infrared exponents are proposed for d=3 in many articles (including Peláez’s) and no concrete comparison to those values is done. I think that this is very easy to be incorporated in the present manuscript.

We have added the reference and others. As with respect to a logarithmic behavior instead/on top of the observed power-law: Given the statistical accuracy of the data, an additional logarithmic deviation from a pure power-law is not possible to discern. We added a corresponding appendix with details. We have also added a statement of commensurability of the fitted exponent with those proposed in the literature.

2) In my impression, the first equation (without number, before (1)) defines a vertex which is not properly renormalized. At odds with the ghost-gluon vertex that a similar > expression gives a vertex which is already renormalized, for the 3-gluon vertex the considered expression would have in d=4 an ultraviolet multiplicative divergence. This is > not problematic for the purposes of the present article but should be pointed out in the text in order to avoid any confusion.

Within the statistical accuracy of the data the corresponding renormalization factors are equal to one. We have added a corresponding comment. The quantity in equation (1) is not intended to be the renormalized one, just as with corresponding other quantities. We also added a comment to this effect.

3) Two lines after equation (5), the authors mention perturbative results to compare to but they do not include any reference. A reference would be welcome.

This has been added.

4) When estimating power-law exponents one usually requires at least a range of two orders of magnitude of variation of the independent variable. Here the authors present figures as 2 and 6 with only one order of magnitude in the power-law range of momenta. In my opinion, this short range of momenta permits to suggest that a power law is present but does not allow to calculate the exponents beyond the order of magnitude. In particular, in my opinion, one cannot estimate with such a narrow interval of momenta exponents with the number of digits that the authors claim. My opinion seems to be confirmed by the figures 3 and 7. Moreover, the authors observe an important volume dependence of their results. In my opinion, the precision of the exponents is overestimated by at least an order of magnitude. I think a rephrasing and a much more careful wording on this point would be welcome. On the same token, the authors suggest that the power-law exponents for the non-tree-level tensor structures are the same than those for the tree-level one. In my opinion, a much careful wording on this point would be welcome because I do not see how with these data one can claim that (beyond a qualitative assertion).

The errors are statistically only, and obtained in the way described. We added a comment on that the error does not include systematic effects beyond the ones described in the text.

Of course, a larger range of momenta would be desirable. However, doubling the range at the same lattice spacing, would increase computing time, even in three dimensions by an order of magnitude, and the memory demands likewise, beyond what is currently at our disposal, or likely feasible.

However, to give a glimpse at very small momenta, we added statistics and additional simulations on even coarser lattices in three dimensions. As they turn out to be not strongly affected by discretization artifacts, as is discussed in detail in an a new appendix, they can be used for that purpose. They do not show any qualitative change. Four dimensions, like in figure 6, were anyhow only seen as a consistency check.

5) In figure 5 one observes a very important noise in the ultraviolet. This has been observed for the same quantity in previous lattice simulations. In my opinion, the reason for such large fluctuations should be discussed in the manuscript or at least the point commented. It may happen that the authors do not know the reason that provokes such a noisy curve but at least they must mention a list of possible origins and discuss more properly the point (that is just mentioned in the manuscript).

The reason is clearly the usual problem that the noise increases exponentially with the number of involved operators (even without disconnected pieces as in the present case). We have added a corresponding comment. There is unfortunately no possibility to eliminate it, except for larger statistics, which we did within our possibilities. Usual noise reduction techniques like smearing would invariably alter the high-momentum behavior.

6) I do not understand the first sentence of page 5.

We are not quite sure what the concrete problem is with the sentence in question, as its meaning is discussed subsequently. We will be happy to modify it given a more concrete description of the problem with it.

Report 2

1) Apart from showing evidence in favor of the zero crossing, the authors try to characterize the infrared behavior of the correlation function. They fit their lattice results with a power law, eq. (4), and try to extract the associated exponent b. I must say that I am not convinced by the determination of the error bars, which seem underestimated. For instance, in section 3-1, the authors give a determination of the critical exponent b as 1.76 with an error of a few percents. But the lattice data show an approximate linear behavior in the log-log plot on only 1 order of magnitude (see fig 2). On such a small range of momenta, the determination of the exponent with this precision would need to quantify the effect of corrections to scaling and this is not addressed by the authors. (A small comment: on the top left plot of fig. 2 The fit seems to be always above the lattice points. This looks strange.)

As this address the same issue as referee 1 in question 4, we refer to the answer there.

Concerning the top-left in figure 2: As noted in the text, we did not adapt the prefactor by fitting for this different momentum configuration (90 as opposed to 180 degrees) to highlight by the mismatch with the data that the prefactor shows an angular dependency, while the power-law remains essentially the same. As this has obviously been a source of confusion,w e have eliminated this.

2) The authors seem to ignore the calculation of Pelaez et al (arXiv:1310.2594) where the 3-point correlation function was computed. These authors proposed a simple understanding for the zero crossing and give a prediction for the singular behavior of the correlation function, both in 3 and 4 dimensions. The authors should compare their findings with earlier predictions for their exponent b.

As this address the same issue as referee 1 in question 1, we refer to the answer there.

3) The article is in general well written. I should mention though that: -below eq 3, I didn't understand what the author meant in the sentence: "Note that for SU(2) ... suppressed"; -in the next sentence it is not clear to me what "all of them" corresponds to; -next paragraph, the first sentence should be rewritten.

Corrected.

There have also be minor corrections and wording throughout the document.

---

## Editorial Decision

published